# Ethnic disparities in medication adherence? A systematic review examining the association between ethnicity and antidiabetic medication adherence

Rayah Asiri[1,2], Adam Todd[1], Anna Robinson-Barella[1], Andy Husband[1] *

**1** School of Pharmacy, Newcastle University, Newcastle upon Tyne, United Kingdom, **2** School of Pharmacy, King Khalid University, Abha, Saudi Arabia

☯ These authors contributed equally to this work.
* andy.husband@newcastle.ac.uk

## Abstract

**Data Availability Statement:** All relevant data are within the paper and its Supporting Information files.

### Objectives

Adherence to prescribed medication is an essential component of diabetes management to obtain optimal outcomes. Understanding the relationship between medication adherence and ethnicity is key to optimising treatment for all people with different chronic illnesses, including those with diabetes. The aim of this review is to examine whether the adherence to antidiabetic medications differed by ethnicity among people with diabetes.

### Methods

A systematic review was conducted of studies reporting adherence to antidiabetic medication amongst people from different ethnic groups. MEDLINE, Embase, CINAHL, and PsycINFO were searched from their inception to June 2022 for quantitative studies with a specific focus on studies assessing adherence to antidiabetic medications (PROSPERO: CRD42021278392). The Joanna Briggs Institute critical appraisal checklist and a second checklist designed for studies using retrospective databases were used to assess study quality. A narrative synthesis approach was used to summarize the results based on the medication adherence measures.

### Results

Of 17,410 citations screened, 41 studies that included observational retrospective database research and cross-sectional studies were selected, each of which involved diverse ethnic groups from different settings. This review identified a difference in the adherence to antidiabetic medications by ethnicity in 38 studies, despite adjustment for several confounding variables that may otherwise explain these differences.

**Funding:** This work was supported by the Saudi Arabian Cultural Bureau in the United Kingdom and King Khalid University in Saudi Arabia. The funders had no role in study design, data collection and analysis, decision to publish, or preparation of the manuscript.

**Competing interests:** The authors have declared that no competing interests exist.

## Conclusion

This review revealed that adherence to antidiabetic medication differed by ethnicity. Further research is needed to explore the ethnicity-related factors that may provide an explanation for these disparities.

## Introduction

Diabetes is a common and clinically significant metabolic disorder, and, in recent decades, has become a major medical burden, reaching the point of a global pandemic [1, 2]. In 2019, an estimated 463 million people worldwide are thought to have diabetes [1], and this number is set to increase in the future: by 2045, the prevalence is estimated to reach 700 million globally [1, 3]. Furthermore, people with diabetes are at increased risk of developing microvascular (*e.g.*, ischemic heart disease and cerebrovascular disease) and macrovascular complications (*e.g.*, nephropathy and retinopathy) as well as having an increased risk of all-cause mortality [4–6]. Accordingly, glycemic control on a long-term basis is critical to diabetes management and good control has been shown to reduce diabetic complications [7, 8]. In conjunction with lifestyle management, such as diet and physical activity, improving adherence to antidiabetic medication is an important aspect of management to delay or prevent the associated complications [9]. Improving medication adherence is considered an integral part of any diabetes management plan for patients [10]. Adherence is defined as the extent to which patients follow their prescriber's instructions when taking medicines [11]. It can be measured using direct or indirect measurements [12]. Direct measurements include measuring the concentration of the medicines, their metabolite and detection of a biological marker given with drug in bodily fluids such as blood or urine while observing the patient's administration of their medications [12]. Indirect methods are the more popular in adherence research as the direct method is costly and challenging to perform [12]. It can be classified according to the World Health Organization into objective and subjective measures [13]. Subjective measurements include methods needing either patient self-report or health care professional evaluations of adherence to prescribed medicines [13]. On the other hand, objective measurements involve counting of pills, electronic monitoring, analysis of the secondary database [13]. In 2003, the World Health Organization (WHO) reported that 'increasing the effectiveness of adherence interventions may have a far greater impact on the health of the population than any improvement in specific medical treatment' [14]. Improving adherence to diabetic medication is associated with improved glycemic control, reduced risk of hospitalization and mortality, as well as lower healthcare expenditure [15, 16]. There are, however, many factors that affect adherence to diabetes medication, including disease-related (*e.g.*, the course of the illness and presence of other comorbid conditions) and treatment-related factors (complex therapy, adverse effects, and polypharmacy) [14, 17–19]. Moreover, such factors can also be influenced by information accessibility and transparency, community support, beliefs and attitudes towards a healthy lifestyle, the patient–healthcare provider relationship, mental health, and self-efficacy [20]. Furthermore, patient characteristics, such as socioeconomic background and cultural beliefs, also have potential to influence how people adhere to their medicines [14]. One such patient factor that has seldom been studied in this context is ethnicity. Given there is increased incidence of diabetes in certain ethnic groups [21], it is important to establish if, or how, adherence to diabetes medication varies by ethnic groups. This systematic review, therefore, aimed to explore whether medication adherence to antidiabetic medications in People with diabetes varied by ethnicity.

## Methods

This systematic review was registered with PROSPERO (CRD42021278392) and conducted with the Preferred Reporting Items for Systematic Reviews and Meta-Analyses (PRISMA) guidelines (S1 and S2 Tables) [22].

### Search strategy and study selection

A systematic and comprehensive literature search was conducted in the following electronic databases: MEDLINE, Embase, CINAHL, and PsycINFO; from inception to June 2022. No search restrictions were applied to the language or date of publication. Quantitative observational studies and the quantitative sections of studies with mixed methods, examining adherence to antidiabetic medications according to ethnicity were eligible for inclusion.

The inclusion criteria of this review were conceptualized by the PICOS (Population, Intervention, Comparison, Outcome and Study Design) framework:

- *Population*: people from any ethnic groups with type 1 or type 2 diabetes mellitus.

- *Intervention*: insulin and/or other antidiabetic medication, including sulfonylureas, biguanides, meglitinide, thiazolidinedione, dipeptidyl peptidase-4 inhibitors, sodium- glucose cotransporter inhibitors, α-glucosidase inhibitors, and glucagon-like peptide-1 receptors agonists.

- *Comparison*: people from any other ethnic groups with type 1 or type 2 diabetes mellitus.

- *Outcome*: Medication adherence or persistence, measured by any medication adherence measure (measures based on the data of electronic databases, self-reported measures, pill counts, electronic monitoring, and biomedical measures), reported according to ethnicity.

- *Study design*: quantitative observational and mixed method studies reporting on adherence by ethnicity.

The studies were required to measure adherence to antidiabetic therapy in more than one ethnic group. Any study only reporting adherence for one ethnic group only was excluded, as were studies that evaluated adherence to all medications used to control diabetes and other comorbid conditions. Qualitative studies, reviews, editorials, conference abstracts, and intervention studies were also excluded from the review.

To complement the database searches, a grey literature search was undertaken. The reference lists of all included studies were hand-searched for relevant papers not retrieved in the database search. The search strategy is described in (S1 Text). The initial screening of titles and abstracts was undertaken by one reviewer (RA), while full-text article screening was conducted by (RA) and checked in full by (AT or AKH). Any disagreement was resolved through discussion and consensus.

### Data extraction and quality appraisal

One reviewer (RA) undertook the data extraction and was checked by another (AR). Data from each study in respect of intervention, population, outcomes, key findings, and study limitations were compiled *via* a customised data extraction form (S5 Table). Three reviewers (RA, AT, AKH) undertook quality assessment using two relevant tools: the Joanna Briggs Institute's critical appraisal tools for cross-sectional studies [23], and a checklist for medication adherence and persistence studies, using retrospective databases [24], as no validated quality assessment tool was suitable for all observational studies.

### Analysis and synthesis

Due to the heterogeneity of ethnic groups, and measures used for medication adherence across the included studies, meta-analysis to combine individual study results was not possible. A narrative synthesis approach was employed to synthesize the data [25]. The influence of ethnicity on adherence to antidiabetic therapy is described based on medication adherence measures used across the included studies: measuring adherence using electronic databases, self-reported measures, and pill counts.

### A note on reporting of the ethnicity data

While some included studies use the term 'race', we prefer the term 'ethnicity', which is defined in this review according to Senior and Bhopal 'implies one or more of the following: shared origins or social background; shared culture and traditions that are distinctive, maintained between generations, and lead to a sense of identity and group; and a common language or religious tradition [26].' The term of 'ethnicity' is preferred as it encompasses cultural and geographical characteristics which will allows to take more confounders into account that are more likely to account for differences in medication adherence. For the included studies, we used labels provided by the authors of the original studies.

## Results

### Literature search and study characteristics

The initial electronic database searches identified 18,107 citations, and another 22 publications were found from reference lists, citation, and a grey literature search. Following the removal of duplicates, 17,410 studies were reviewed for eligibility based on the title and abstract; from this, 164 studies were selected for a full-text review. After full text screening, 41 studies were included in this systematic review (Fig 1).

All articles were published between 2005 and 2022 and were conducted in nine countries: United States (n = 26) [27–52], New Zealand (n = 3) [53–55], United Kingdom (n = 2) [56, 57], Canada (n = 1) [58], Malaysia (n = 4) [59–62], Singapore(n = 2) [63, 64], Brazil (n = 1) [65], Qatar (n = 1) [66] and United Arab Emirates (n = 1) [67]. The majority of included studies were retrospective database studies [27–31, 33–51, 53–58, 60, 63, 64], while the remaining were cross-sectional studies [30, 52, 59, 61, 62, 65–67]. The sample sizes ranged from 57 [60] to 1,888,682 patients [39], and the mean age of patients ranged from 14 to 71 years. Ethnicity was reported across studies as: ethnicity in (14 studies), race/ethnicity in (16 studies), and race in (12 studies). The majority of studies examined adherence to antidiabetic medication among people with type 2 diabetes mellitus; only two studies examined adherence to antidiabetic medications for people with type 1 diabetes mellitus [60, 65]. Thirty studies used medication adherence measures based on the electronic database data: medication possession ratio, the proportion of days covered, or other measures centred on a pharmacy database. Of the remaining nine studies, Two used pill count to measure adherence [41, 52], while nine used self-reported measures of adherence [32, 49, 59, 61, 62, 64–67]. Study characteristics are detailed in (Table 1).

### Quality assessment

Two quality assessment tools were used in this review based on the study design. The first tool, a checklist for medication adherence and persistence studies using retrospective databases, was used for the 33 retrospective database studies [24]. All study titles were descriptive and reflective of the study purpose, and all abstracts provided a short description of the work. The

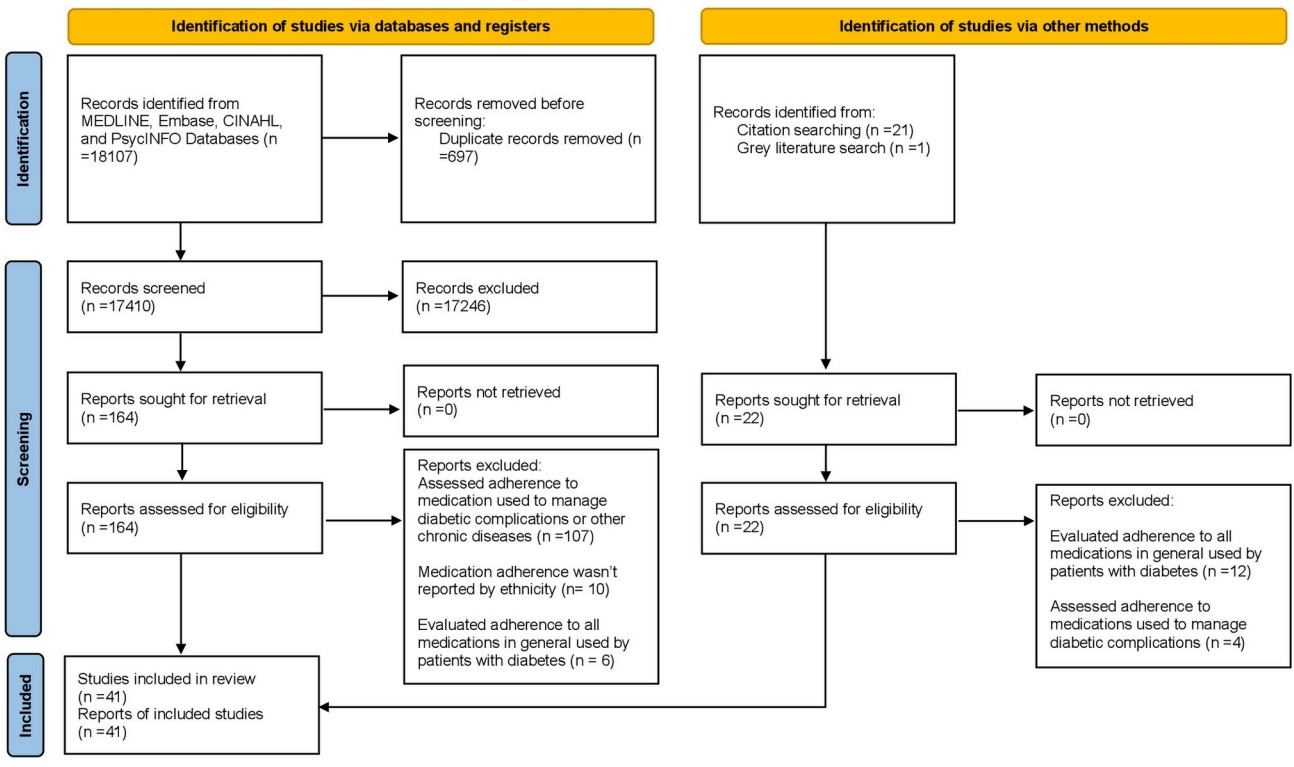

**Fig 1. PRISMA flow diagram of included studies.**

rationale and scientific background were reported in most studies except for five, which did not clearly state their objectives [28, 37, 44, 47, 60]. Most of the articles clearly stated the study design that matched the objectives; only eight did not do so [38, 42, 49, 53–55, 57, 64]. The data sources were clearly defined in all studies; however, three papers did not state the time frame for data [28, 49, 63]. Of the 33 studies, 29 clearly stated the eligibility criteria, but 16 did not explain the rationale for these criteria [27, 28, 30, 36, 41, 47, 49–51, 54, 55, 57, 58, 60, 64]. The method of medication adherence measurement was clearly described in the majority of studies, but, for some studies, the method of handling the switching of drugs within and between therapeutic classes was not appropriately explained [28, 29, 36, 40, 44–48, 50, 51, 55, 57, 58, 63, 64]. All studies apart from three [42, 43, 55], controlled the confounders, while all studies clearly presented their results and discussed the research implications (S3 Table).

The quality of the eight remaining cross-sectional studies was assessed using the Joanna Briggs Institute's critical appraisal tools for cross-sectional studies [23]. All studies clearly defined the inclusion criteria, study subjects, and settings, but two did not report on strategies to deal with confounding factors [32, 61]. All studies, but one, used a valid and reliable measure of medication adherence [65] (S4 Table).

### Narrative synthesis of the influence of ethnicity on adherence to antidiabetic medication by method of measurement

Studies were described below thematically according to medication adherence measures using a narrative synthesis approach [25].

**Table 1. Study characteristics.**

| Study (author/year) | Country | Study Design | Data source | Population (number, ethnicity data) | Antidiabetic medication | Medication adherence measure | Key finding |
|---|---|---|---|---|---|---|---|
| [54] | New Zealand | Retrospective database | Data collected from 10 general practices in the Waikato region. | N = 1595 New Zealand European. Maori. Asian. Pasifika. Others. | Metformin | MPR | Māori and Pasifika patients had poorer metformin adherence than patients of other ethnicities. After adjusted analysis, the difference remained Māori compared to New Zealand Europeans (OR 0.41; 95% CI 0.30–0.56; P<0.001). |
| [37] | US | Retrospective database | Data from Kaiser Permanente Northern California. | N = 9923 White. Black. Hispanic. Asian. Other. | Insulin and/or OADs | PDC | Non-Hispanic Black (ARR, 1.52; 95% CI, 1.32–1.74), Hispanic (ARR, 1.28; 95% CI, 1.15–1.43), and Asian (ARR, 1.14; 95% CI, 1.01–1.29) women were more likely than non-Hispanic White women to be nonadherent. |
| [41] | US | Retrospective database | Data from DPP randomized clinical trial. | N = 30445 White. Black. Hispanic. Other. | Metformin | Pill count | Odds of adherence was significantly lower for Black participants (HR 0.44; 95% CI 0.29–0.66; P<0.001) compared to Whites. |
| [53] | New Zealand | Retrospective database | The New Zealand Ministry of Health's national data collections. | N = 90,530 Māori Pacific European Asian Indian Middle Eastern/ Latin American/ African other and Unknown | Metformin | MPR | After adjusting for all the person- and health-related factors, the annual MPR for Māori (MPR -0.08, 95% CI -0.08 to 0.07) and Pacific (MPR-0.09, 95% CI -0.10 to 0.08) peoples were lower than Europeans. |
| [59] | Malaysia | Cross-sectional | - | N = 232 Malay Chinese Indian others | Insulin and/or OADs | Validated Medication Compliance Questionnaire | Malay patients had 1.43 times higher odds of nonadherence to medications than Chinese and Indian patients (95% CI 1.32–1.74; p = 0.031). |
| [55] | New Zealand | Retrospective database | Pharmaceutical Collection Database maintained by the New Zealand's Ministry of Health. | N = 7138 New Zealand (NZ) European. Other European NZ Māori Indian | OHAs | PDC | Adherence was significantly higher in the NZ European ethnic group (PDC 64.4 vs. 9.1,20.1, and 2.8; P < 0.001). Non-adherence was significantly higher in patients in New Zealand Māori patients (PDC 33.3 vs. 48.3,7.6, and 4.8; P < 0.001). |
| [38] | US | Retrospective database | Longitudinal claims data from a large US-based insurance provider | N = 56720 White Asian Black Hispanic | OADs | PDC | Average adherence rates of blacks and Hispanics were 4.8 to 6.5 percentage points lower than whites (p <0.01). |
| [61] | Malaysia | Cross-sectional | - | N = 497 Malay. Indian. Chinese | OADs | MMAS | There was statistically significant association between ethnicity and medication adherence ($X^2$ = 10.660; P = 0.031). The Indians had higher medication adherence compared with the Malays and the Chinese |

(*Continued*)

**Table 1.** (Continued)

| Study (author/ year) | Country | Study Design | Data source | Population (number, ethnicity data) | Antidiabetic medication | Medication adherence measure | Key finding |
|---|---|---|---|---|---|---|---|
| [31] | US | Retrospective database | The Veterans Affairs electronic health records & outpatient pharmacy | N = 148,544 White African American Asian Native American Hawaiian/Pacific Islander | OADs | PDC | Except for Asians, all other minorities were less likely to be adherent compared with whites (OR = 0.62; 95% CI = 0.605–0.645). |
| [68] | UK | Retrospective database | Royal College of General Practitioners Research and Surveillance Centre database | N = 60327 White Asian Black Mixed Other | OADs | CMG | Non-white ethnicity associated with reduced medication persistence compared to Whites (HR 1.53; 95%CI 1.47–1.59; P<0.001) |
| [66] | Qatar | Cross-sectional | - | N = 260 Arab Asian Other | Insulin and/or OADs | ARMS-D | Ethnicity was an independent predictor of adhere score (Beta -1.731; 95% CI -3.153–0.309; p = 0.017). Arab ethnicity was likely to be non-adherent compared to Asian. |
| [30] | US | Retrospective database | Administrative claims data Hawaii | N = 5163 Japanese Native Hawaiians Whites Filipinos Chinese Mixed race and other race. | OADs | PDC | After adjustment, Filipinos [OR = 0.58, 95%CI (0.45,0.74)], Native Hawaiians [OR = 0.74, 95%CI(0.56,0.98)], and people of other race [OR = 0.67, 95%CI (0.55,0.82)] were significantly less adherent to anti-diabetic medications than Japanese |
| [64] | Singapore | Retrospective database | Public primary care clinic | N = 5163 Chinese Malay Indian Other | OADs | MARS-5 | Compared to Indian, patients of Chinese ethnicity (OR 2.80; 95% CI 1.53–5.15; P<0.01) had low adherence to OADs. |
| [60] | Malaysia | Retrospective database | Clinic Paediatric at University Kembangan Malaysia Medical Centre | N = 57 Malay Chinese Indian Others | Insulin | MPR and glycated haemoglobin | Race was not associated with adherence(p = 1.000) |
| [42] | US | Retrospective database | Administrative claims data from a large health insurance company in Hawaii | N = 30445 Japanese. Filipino. Chinese. Caucasian. Native Hawaiian. Other Pacific Islander. | Insulin and/or OADs | MPR | Native Hawaiians, Other Pacific Islanders, and Filipinos had significantly lower adherence rates for each medication compared to other groups. |
| [63] | Singapore | Retrospective database | Multicentre data of the National Healthcare Group | N = 2463 Chinese Malay Indian Others | Insulin and/or OADs | PDC | Indian ethnicity (OR 0.59; 95% CI 0.48–0.73; P<0.001) was associated with poorer adherence compared with other ethnic groups in Singapore |
| [28] | US | Retrospective database | Kaiser Permanente Northern California Diabetes Registry | N = 30,838 White. Hispanic/Latino. African American. Asian. | Insulin and/or OADs | CMG | Latinos had greater non-adherence than white patients (p<0.05) |

(*Continued*)

**Table 1.** (Continued)

| Study (author/year) | Country | Study Design | Data source | Population (number, ethnicity data) | Antidiabetic medication | Medication adherence measure | Key finding |
|---|---|---|---|---|---|---|---|
| [35] | US | Retrospective database | Data from Kaiser Permanente Southern California | N = 23612 Non-Hispanic White Black Hispanic Asian or Pacific Islander Other/Unknown | OHAs | CMG | Black race (RR 1.28;95% CI 1.17–1.41), Hispanic ethnicity (RR 1.11; 95% CI 1.03–1.20) were associated with discontinuation of one or dual OHAs. |
| [36] | US | Retrospective database | MarketScan Multi-State Medicaid Database | N = 1529 Caucasians African Americans Other races | OADs | MPR | African Americans (OR 0.71; 95% CI 0.55–0.93; P<0.05) had significantly lower MPR than Caucasian patients. |
| [65] | Brazil | Cross-sectional | - | N = 1698 Caucasian Non-Caucasian | Insulin | Self-reported scales | Adherence to insulin was not related to ethnicity. |
| [43] | US | Retrospective database | Pennsylvania Medicaid administrative claims data | N = 16,256 White. Black. Hispanic. Other/unknown. | OHAs | PDC | Compared to perfect adherers, enrolees who discontinued oral hypoglycaemics were non-white groups (43.3–57.3% vs. 37.3%; P<0.0001) |
| [67] | UAE | Cross-sectional | - | N = 446 Arab Emarati. Arab Non-Emarati Asian. | Insulin and/or OADs | MMAS | Adherence levels differed significantly between ethnic groups (p<0.001) with the lowest adherence levels for Emirati patients (81.6%) followed by Arab Non-Emirati (47.1%) and Asians (15.4%). |
| [50] | US | Retrospective database | Group Health ooperative in Washington State and parts of Idaho | N = 509 White African American American Indian / Alaska Native. Asian / Pacific Islander Not Hispanic Hispanic | Insulin and/or OADs | MPR | Non-Hispanic were more likely to be non-adherent to diabetes medications compared to Hispanic. |
| [32] | US | Cross-sectional | - | N = 7239 White African American Hispanic Asian American American Indian. | Insulin and/or antidiabetic drugs | MMAS | The White group had higher medication adherence (50.3%; P<0.001) compared to all groups except the American Indian group. |
| [58] | Canada | Retrospective database | Health administrative and pharmacy data from British Columbia | N = 167243 White South Asian Chinese | Insulin and/or OADs | PDC | South-Asian (OR 0.44; 95% CI 0.41–0.47; P<0.0001) and Chinese people (OR 0.87; 95% CI 0.82–0.93; P<0.0001) had significantly lower adherence compared with white people. |
| [47] | US | Retrospective database | A Hawaiian health plan. | N = 43445 White Japanese Chinese Filipino Native Hawaiian Other Pacific Islander | Insulin and/or OADs | PDC | Filipinos, Native Hawaiians, and other Pacific Islanders were significantly less adherent than Whites(p<0.05). |

(*Continued*)

**Table 1.** (Continued)

| Study (author/year) | Country | Study Design | Data source | Population (number, ethnicity data) | Antidiabetic medication | Medication adherence measure | Key finding |
|---|---|---|---|---|---|---|---|
| [57] | UK | Retrospective database | Aggregated prescribing data of 76 General Practice surgeries within the Heart of Birmingham teaching Primary Care Trust | N = 23687 White Asian Black Mixed Other | OADs | MPR | There was a statistically significant variations in adherence between the major ethnic groupings. (MPR range from 86.8% for White groups to 58.3% for Black groups) |
| [48] | US | Retrospective database | A large, Midwestern, integrated health system | N = 31636 White Black Asian Hispanic American Indian | Antidiabetic drugs | MPR | Non-White patients (OR 1.66; 95% CI 1.41–1.95; P<0.001) have statistically significant lower adherence compared to the White |
| [40] | US | Retrospective database | Data from Kaiser Permanente Colorado | N = 12,061 Non-Hispanic White Hispanic. Other. | Antidiabetic drugs | PDC | Patients of Hispanic ethnicity (OR 1.33; 95% CI 0.84–2.10) were more likely to be primarily non-adherent compared to White. |
| [51] | US | Retrospective database | Texas Medicaid prescription claims data | N = 3109 White Black Hispanic Other | OADs | MPR | Race was statistically significantly related to adherence. The adherence was two times higher for whites compared to. Hispanics (OR 2.02; 95% CI, 1.54–2.66; P<0.0001). |
| [27] | US | Retrospective database | The National Patient Care &Pharmacy Benefits Management database | N = 690,968 NHW NHB Hispanic Other | Insulin and/or OHAs | MPR | Relative to non-Hispanic whites, MPR was significantly (6.07%) lower in non-Hispanic blacks, and (1.76%) Hispanics. |
| [44] | US | Retrospective database | Indiana Network of Patient Care | N = 3,976 White. African American | OADs | PDC | African American race were a significant risk factor for non-adherence compared to White race (OR 0.61; 95% CI 0.50–0.73; P<0.001) |
| [45] | US | Retrospective database | Veterans Administration datasets | N = 11,272 NHW NHB Hispanic Other | Insulin and/or OHAs | MPR | NHB had statistically significantly lower MPR compared to NHW. |
| [49] | US | Retrospective database | Two primary care clinics and two diabetes specialty clinics | N = 383 White African American | Antidiabetic drugs | SDSCA questionnaire | African American race was significantly associated with less medication adherence (r = −0.10, p<0.05) |
| [39] | US | Retrospective database | Administrative claims data primarily pharmacy claims data | N = 1,888,682 White Black Hispanic Others | OHAs | PDC | Black (OR 1.39; 95% CI 1.38–1.41; P<0.001) and Hispanic patients (OR 1.37; 95% CI 1.35–1.39; P<0.001) more likely than whites to be nonadherent. |
| [29] | US | Retrospective database | Harvard Vanguard Medical Associates electronic medical record system | N = 1906 Black White | Insulin and/or OADs | Monthly average rate of adherence | Blacks had statistically significantly lower MPR compared to Whites (IRR 079; 95% CI 0.63–0.99; P<0.05). |
| [33] | US | Retrospective database | Harvard Vanguard Medical Associates | N = 1806 Black White | OADs | Refill-based medication adherence | Black patients had a significant lower average of medication adherence compared to whites. (71.7% vs. 77.6%; P<0.0001) |

(*Continued*)

**Table 1.** (*Continued*)

| Study (author/ year) | Country | Study Design | Data source | Population (number, ethnicity data) | Antidiabetic medication | Medication adherence measure | Key finding |
|---|---|---|---|---|---|---|---|
| [46] | US | Retrospective database | Data from the North Carolina Medicaid program | N = 17,685 African American. White | OADs | MPR | The adherence rate of African American patients was significantly lower by 12% than whites (p<0.05) |
| [34] | US | Retrospective database | A large health care plan in Hawaii | N = 20685 Japanese Chinese Whites Hawaiians Filipinos other | OADs | MPR | Compared to Whites, Japanese patients (OR 1.20; 95% CI 1.0–1.30) were the most likely to be adherent, followed by Chinese (OR 1.0; 95% CI 0.89–1.20), Hawaiians (OR 0.89; 95% CI 0.77–1.0), and Filipinos (OR 0.78; 95% CI 0.68–0.90) |
| [52] | US | Cross-sectional | - | N = 381 Hispanic Non-Hispanic Black Non- Hispanic White | Insulin and/or OADs | Unannounced pill count | Non-Hispanic Blacks were 5 and 6 times more likely to be low-adherent to OADs than Hispanics and non-Hispanic whites, respectively (95% CI: 2.28–11.90, p < 0.001) (95% CI: 2.12–15.49, p = 0.001) |
| [62] | Malaysia | Cross-sectional | - | N = 249 Chinese Indian Malay | Insulin | IAQDM | There was no significant association between ethnicity and adherence to insulin therapy. |

MPR: medication possession ratio, PDC: proportion of days covered, MMAS: Morisky Medication Adherence Scale, CMG: continuous single-interval measure of medication gaps, ARMS-D: Adherence to Refill and Medications Scale in Diabetes, MARS-5: five-question Medication Adherence Rating Scale, SDSCA: the Summary of Diabetes Self-Care Activities questionnaire, IAQDM: Insulin Adherence Questionnaire for Diabetes Mellitus, OHAs: oral hypoglycaemic agents, OADs: oral antidiabetic drugs, NHW: Non-Hispanic Whites, NHB: Non-Hispanic Blacks, US: United State, UK: United Kingdom, OR: odd ratio, CI: confidence interval, ARR: adjusted risk ratio, P: p value, HR: hazard ratio, RR: relative risk, r:correlation coefficient, IRR: incidence rate ratio.

**Studies measured adherence using electronic databases.** A variety of measures using pharmacy refill data were used to assess adherence to antidiabetic medications among different ethnic groups, including the MPR [27, 30, 36, 42, 45, 46, 48, 50, 51, 53, 54, 57, 60, 64], proportion of days covered [31, 37–40, 43, 44, 47, 55, 58, 63], and other measures, such as medication refill adherence and continuous single- or multiple-interval measures of medication gaps [28, 29, 33, 35, 56].

*Medication possession ratio*. Fourteen studies used the MPR to evaluate medication adherence for people with diabetes from different ethnic groups. Three studies assessed adherence to antidiabetic medications among the following ethnic groups: Japanese, Hawaiians, Whites, Filipinos, Chinese, mixed races, and other races [30, 42, 64]. The reported adherence rates varied by ethnicity with people of Japanese ethnicity most likely to be adherent to antidiabetic medications, followed by Whites, Chinese, Hawaiians, Filipinos, and other ethnic groups; this difference was statistically significant in two studies [30, 64]. The difference between people from the following ethnicities: Whites, African Americans, Hispanics, Asians, American Indians, and others in adherence to antidiabetic medications was also reported in seven studies [27, 36, 45, 46, 48, 50, 51]. People with a Non-White ethnicity had a statistically significant lower MPR compared to people with White ethnicity across these studies. Two other studies examined adherence to metformin among ethnic groups in New Zealand and revealed that people from Māori and Pacific ethnic groups had poorer adherence than people from the European group [53, 54]. Langley and colleagues also reported statistically significant

variations in adherence to oral antidiabetic medications among different ethnic groups in the United Kingdom, with highest adherence rates reported for people from Irish ethnic groups, while lowest adherence rates were reported for people from African ethnic groups [57]. Only one study assessed adherence to insulin therapy for type 1 diabetes; across three Asian ethnic groups–Malay, Chinese, and Indian–ethnicity was not associated with adherence [60].

*Proportion of days covered.* Eleven studies used the proportion of days covered to assess adherence to antidiabetic medications by people of different ethnic groups. Of these, seven studies indicated that people from Black, Hispanic, and Asian ethnicity were more likely to be non-adherent to antidiabetic medications than people of White ethnicity, and these findings were statistically significant in four studies [37–40, 43, 44]. The adherence levels also differed among people with White, Chinese, Black, Hawaiian, Japanese, Filipino, and Native American ethnicities [31, 47, 58]. In contrast to the studies using the MPR, one study found that the medication adherence of people from Chinese and Japanese ethnic groups did not differ significantly from people from White ethnicity; however, members of other ethnic groups were found to be significantly less adherent compared to people from White ethnic groups [47]. Additionally, significant differences in adherence to antidiabetic agents were reported among three Asian ethnic groups in a study conducted in Singapore, in which people from Indian ethnic groups had the lowest adherence rates compared to people from Malay or Chinese ethnic groups [63]. Lastly, Kharjul and colleagues reported a significantly higher rate of non-adherence among people from Māori ethnicity compared to other ethnic groups in New Zealand, which is consistent with the findings of studies assessing the same groups, using the MPR as a measure of adherence [55].

*Other measures (medication refill adherence and continuous single- or multiple-interval measures of medication gaps).* Two studies, estimating adherence based on pharmacy refill data, reported a statistically significant difference in adherence to antidiabetic medication between people from Black and White ethnicities [29, 33]. Medication persistence was also assessed in three studies of different ethnic groups using a continuous single-interval measure of medication gaps. These studies showed that non-White ethnicity was associated with lower medication persistence when compared to White ethnicity [28, 35, 56].

**Study measured adherence using pill count.** Two studies used pill counts to assess adherence to oral antidiabetic medications for three ethnic groups: Whites, Blacks, and Hispanics. The authors found that the adherence varied significantly by ethnicity, and the odds of adherence were lower for people with Black ethnicity than those of White and Hispanic ethnicity [41, 52].

**Studies measured adherence using self-report measures of adherence.** Eight studies assessed adherence to antidiabetic medications using a variety of self-report measures, including the five-question Medication Adherence Rating Scale [64], Morisky Medication Adherence Scale [32, 61, 67], validated Medication Compliance Questionnaire [59], Insulin Adherence Questionnaire for Diabetes Mellitus [62], Adherence to Refill and Medications Scale in Diabetes [66], medication adherence subscale of the Summary of Diabetes Self-Care Activities questionnaire [49], and self-reported scales [65]. In three of these studies, statistically significant differences in adherence were found among three Asian ethnic groups: Chinese, Malays, and Indians [59, 61, 64]. Lopez and colleagues [32] and Osborn and colleagues [49] demonstrated a significant association between increased adherence to antidiabetic medications and White ethnicity compared to compared to African American, Hispanic, Asian, and American Indian ethnic groups [32, 49]. A study conducted in the United Arab Emirates also reported significant variation in adherence levels to antidiabetic medications among three ethnic groups (Arabs of the United Arab Emirates, Arabs non-Emirate, and Asians) with the highest level of adherence reported for people with Asian ethnicity and the lowest for people with Arab ethnic

groups [67]. The findings of this study are consistent with another study conducted in Qatar, in which people with diabetes of Arab ethnicity were more likely to be non-adherent to their medication compared to people from Asian ethnicity [66]. Only two study found that adherence to antidiabetic medications did not differ by ethnicity in people with type 1 diabetes [62, 65].

## Discussion

This systematic review aimed to examine the association between ethnicity and adherence to antidiabetic medications in people with type 1 and type 2 diabetes mellitus. This review identified several studies showing that adherence to antidiabetic medications varied according to the ethnic groups. This variation was statistically significant in 34 studies out of 41 included studies and was overall observed between people from ethnic minorities and the majority populations in each specific study setting. Moreover, it is reported when adherence was measured using various measurement methods. The finding of this review supports the work of Peeters and colleagues who reported a direct association between ethnicity and adherence across three studies [20]. This work, however, only focused on oral hypoglycemic medications in type 2 diabetes, and was published in 2011 (and since this time more relevant literature has been published) [20]. Taken together, these findings may help explain some of the ethnic/racial disparities in diabetes outcomes [69–71], as optimum glycemic control is one of the major factors influencing diabetes outcome [72]. Nonetheless, these findings also offer a strong foundation for exploring the ethnicity-related factors that might explain the differences in adherence to antidiabetic medications. Understanding the possible reasons why these differences occur amongst minority ethnic groups will help develop interventions tailored to patients in ethnically diverse populations. Given the emphasis of previous literature emphasizing the need to understand patient-specific factors to design effective adherence interventions, the findings of this work are even more relevant for future research [73].

One interesting finding of this review is that most studies reported statistically significant differences in adherence to antidiabetic medications after controlling for different confounders (*e.g.*, socioeconomic variables) mediating the association between ethnicity and adherence to antidiabetic medications. However, the ethnicity–adherence association in some studies could be attributed to socioeconomic differences among ethnic groups, or other uncontrolled confounders, such as clinical, demographic, and psychological factors, which could mediate the ethnicity- adherence association. In accordance with the present results, one previous study demonstrated that improved health literacy diminished the ethnic differences in adherence to antidiabetic medications [49].

A possible explanation for the differences in adherences to antidiabetic medications among different ethnic groups after controlling confounders mediating adherence-ethnicity associations is related to the socio-cultural background and patients' beliefs and perceptions about treatment, which was shown in a systematic review of 25 cross-sectional studies to be significantly associated with medication adherence in patients with type2 diabetes, asthma, and hypertension [74]. The association may also be explained by the findings reported in several qualitative studies of the barriers of nonadherence to antidiabetic medications and diabetes self-management in some ethnic groups. For example, social stigma and cultural pressure associated with diabetes in people from South Asian background as a barrier for diabetes management [75], the preference of traditional medicines over the use of metformin in people from Māori ethnicity as a cause for suboptimal adherence [76] and feeling frustrated and fearful about taking medicines lifelong in people from African American ethnicity as a reason for nonadherence [77], are all possible explanations of the disparities. Future work could examine

the socio-cultural factors affecting adherence to antidiabetic medications for different ethnic groups.

This review has some limitations. Firstly, it is important to acknowledge the diversity among included studies regarding the settings, designs, measurements of adherence to antidiabetic medications, and inconsistency in reporting ethnicity data. It is therefore not possible to compare across included studies to determine which ethnic group is the most adherent in the context of diabetes management. Secondly, there is no validated quality assessment tool for all observational studies, including retrospective database observational studies and cross-sectional studies. Consequently, using two critical appraisal tools does not allow for a comparison of quality across all studies. Lastly, the included studies primarily used self-reported measures or pharmacy claims data to assess adherence, leading to overestimating adherence or failing to reflect the actual medication-taking behaviour [78]. Therefore, developing a cross-culturally validated adherence measure in future work may help identify and improve adherence issues across different ethnic groups [79]. Notwithstanding these limitations, the results of this review have important implications for optimizing adherence to antidiabetic medications among different ethnic groups. Additionally, the large number of studies that met the eligibility criteria included in this review by using search strategies include broad terms with no restrictions on publication date or language, and an additional grey literature search complemented the original canvass, ensuring the inclusion of all relevant studies (S1 Text).

## Conclusion

This systematic review is one of the first to examine the association between ethnicity and medication adherence in people with type 1 and type 2 diabetes mellitus. The findings show difference among diverse ethnic groups regarding adherence to antidiabetic medications, and these differences persist after controlling for several different demographic, clinical, socioeconomic, and psychological factors. Future research should therefore focus on explaining and understanding why these disparities occur amongst ethnic groups. Understanding the reasons behind these disparities will have important implications for the future care of people with diabetes, including developing possible patient-centred interventions for optimizing adherence to diabetic therapy.

## Supporting information

**S1 Table. PRISMA 2020 checklist.**
(DOCX)

**S2 Table. PRISMA 2020 for abstracts checklist.**
(DOCX)

**S3 Table. Results of the risk of bias assessment of retrospective database studies.**
(DOCX)

**S4 Table. Results of the risk of bias assessment of cross-sectional studies.**
(DOCX)

**S5 Table. Data extraction form.**
(DOCX)

**S1 Text. Database search terms.**
(DOCX)

## Author Contributions

**Conceptualization:** Rayah Asiri, Adam Todd, Andy Husband.

**Data curation:** Rayah Asiri, Anna Robinson-Barella, Andy Husband.

**Formal analysis:** Andy Husband.

**Methodology:** Rayah Asiri, Adam Todd, Anna Robinson-Barella.

**Project administration:** Adam Todd, Andy Husband.

**Supervision:** Adam Todd, Andy Husband.

**Writing – original draft:** Rayah Asiri.

**Writing – review & editing:** Adam Todd, Andy Husband.

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
