## [Decision Letter · Decision Letter 0]

9 Sep 2022

PONE-D-22-18891Ethnic disparities in medication adherence? A systematic review examining the association between ethnicity and antidiabetic medication adherencePLOS ONE

Dear Dr. Husband,

Thank you for submitting your manuscript to PLOS ONE. After careful consideration, we feel that it has merit but does not fully meet PLOS ONE’s publication criteria as it currently stands. Therefore, we invite you to submit a revised version of the manuscript that addresses the points raised during the review process.

We look forward to receiving your revised manuscript.

Kind regards,

Gianluigi Savarese

Academic Editor

PLOS ONE

Journal Requirements:

"This work was supported by the Saudi Arabian Cultural Bureau in the United Kingdom and King Khalid University in Saudi Arabia."

5. We note that this manuscript is a systematic review or meta-analysis; our author guidelines therefore require that you use PRISMA guidance to help improve reporting quality of this type of study. Please upload copies of the completed PRISMA checklist as Supporting Information with a file name “PRISMA checklist”.

Reviewers' comments:

Reviewer's Responses to Questions

**Comments to the Author**

1. Is the manuscript technically sound, and do the data support the conclusions?

Reviewer #1: Yes

Reviewer #2: Yes

2. Has the statistical analysis been performed appropriately and rigorously? 

Reviewer #1: N/A

Reviewer #2: N/A

3. Have the authors made all data underlying the findings in their manuscript fully available?

Reviewer #1: Yes

Reviewer #2: Yes

4. Is the manuscript presented in an intelligible fashion and written in standard English?

Reviewer #1: Yes

Reviewer #2: Yes

5. Review Comments to the Author

Reviewer #1: In this systematic review, Asiri et al aimed to examine whether ethnicity could impact the adherence to antidiabetic medications among people with diabetes. Authors deserve praise for trying to address a relevant clinical issue through an adequate methodology and a well-written manuscript. Unfortunately, the available studies are far from conclusive and their review did not provide a clear novelty. In particular, adherence to therapies is heavily influenced by numerous clinical, social and pharmacological factors, which are rarely comprehensively addressed in observational studies. Indeed, the interpretation of the results of the individual studies varies based on numerous factors, such as country, sample size, type of drugs, number of centers, study design, adherence measures, inclusion and exclusion criteria, clinical and social characteristics of the population, comorbidities, adjusted confounders, etc. The fact that the confounding factors considered in the multivariate models are different in the various studies and that the individual ethnic groups are compared with different groups undermines the reliability of the results presented in a systematic form. Unfortunately, these limitations and concerns are of fundamental nature.

Specific comments:

- Authors should briefly define the various adherence measures.

- How do individual studies (and your review) deal with mixed ethnicity?

- In the results section, the authors often report "other races" and in some cases (eg lines 253-255) a difference of some ethnic groups is reported without reporting the comparison group.

- In the results section, the authors often report "other races”, which should be reported in full. Furthermore, in some cases (e.g. page 16, lines 253-256), a difference of some ethnic groups is reported without mentioning the comparative group.

Reviewer #2: The authors performed a systematic review investigating the adherence to glucose-lowering treatment in patients with diabetes according to different ethnicities.

The results are interesting and this overview is much needed, significantly contributing to current knowledge.

My main concerns are:

- on page 6, the authors correctly state that they prefer the term "ethnicity" to "race". This might be preferable not only for semantic reason, but also because, giving that ethnicity encompasses those cultural and geographical characteristics that are more likely to account for differences in medication adherence. In other words, it is also more technically sound to use the term ethnicity in this context because it ideally allows to take more confounders into account. This should be better reported in the methods section.

- The discussion could might be better elaborated. The main differences between ethnicities are not summarized in a narrative statement in the discussion, although they are thoroughly reported in Table 1.

- were there any important differences between glucose-lowering classes? Is there any factor that makes different ethnicities prefer oral antidiabetics rather than injectables?

- access to diabetes medication in the world cannot be ignored, please refer to the latest WHO report https://www.who.int/publications/i/item/9789241565257

- moreover, despite the lack of studies on type 1 diabetes, it is fundamental to better develop the discussion regarding the challenges of treatment adherence in type 1 vs type 2 patients. Although I understand that this is not the main focus of the paper, discussing adherence in type 1 diabetes cannot disregard the need of injectables, the difference in availability of glucose-monitoring system and a substantially different target population.

Consider for example citing "Hsin O, La Greca AM, Valenzuela J, Moine CT, Delamater A. Adher-ence and glycemic control among Hispanic youth with type 1 diabetes:role of family involvement and acculturation.J Pediatr Psychol. 2010;35(2):156-166" and

- it is certainly true that the cause of these disparities should be better investigated in future research, however that is not the only necessity and is also already partly known. The authors should speculate on possible tools for identifying adherence issues and improving it across different ethnic groups.

6. PLOS authors have the option to publish the peer review history of their article (what does this mean?). If published, this will include your full peer review and any attached files.

Reviewer #1: No

Reviewer #2: No

---

## [Author Response · Author response to Decision Letter 0]

9 Nov 2022

Academic editor comments Authors’ Response

We have verified that all formatting and style requirements have been met.

"This work was supported by the Saudi Arabian Cultural Bureau in the United Kingdom and King Khalid University in Saudi Arabia."

Thanks for the comment. We have changed the funding statement. 

Our updated statement is as follows:

"This work was supported by the Saudi Arabian Cultural Bureau in the United Kingdom and King Khalid University in Saudi Arabia. There was no additional external funding received for this study."

3. We note that you have stated that you will provide repository information for your data at acceptance. Should your manuscript be accepted for publication, we will hold it until you provide the relevant accession numbers or DOIs necessary to access your data. If you wish to make changes to your Data Availability statement, please describe these changes in your cover letter and we will update your Data Availability statement to reflect the information you provide. Thanks for your kind reminders. We revised the data availability statement.

Our updated statement is as follows:

"All data related to this study are included in this published article (and its supplementary information files. "

4. Please include captions for your Supporting Information files at the end of your manuscript, and update any in-text citations to match accordingly. Thank you for the reminder. We included captions for the supporting information files at the end of our manuscript and have updated the in-text citations to match accordingly.

[Page:25, Line:619-623]:

Supporting information

S1 Table. PRISMA 2020 checklist

S1 Text. Database search terms

S2 Table. Results of the risk of bias assessment of retrospective database studies

S3 Table. Results of the risk of bias assessment of cross-sectional studies

S4 Table. Data extraction form”

5. We note that this manuscript is a systematic review or meta-analysis; our author guidelines therefore require that you use PRISMA guidance to help improve reporting quality of this type of study. Please upload copies of the completed PRISMA checklist as Supporting Information with a file name “PRISMA checklist”. 

Thank you for your comment. The PRISMA checklist has already been included as Supporting Information in the supplementary files.

[Supplementary files: Appendix A, S1 Table]

2- Response to reviewer #1

Reviewer #1 comments

1-In this systematic review, Asiri et al aimed to examine whether ethnicity could impact the adherence to antidiabetic medications among people with diabetes. Authors deserve praise for trying to address a relevant clinical issue through an adequate methodology and a well-written manuscript. Unfortunately, the available studies are far from conclusive, and their review did not provide a clear novelty. In particular, adherence to therapies is heavily influenced by numerous clinical, social and pharmacological factors, which are rarely comprehensively addressed in observational studies. Indeed, the interpretation of the results of the individual studies varies based on numerous factors, such as country, sample size, type of drugs, number of centers, study design, adherence measures, inclusion and exclusion criteria, clinical and social characteristics of the population, comorbidities, adjusted confounders, etc. The fact that the confounding factors considered in the multivariate models are different in the various studies and that the individual ethnic groups are compared with different groups undermines the reliability of the results presented in a systematic form. Unfortunately, these limitations and concerns are of fundamental nature. 

Thank you for the comment. We agree that adherence is a complex sociological phenomenon and heavily influenced by numerous clinical, social and pharmacological factors, and we believe that ethnicity is also one of these factors. 

We chose not to pool our data quantitatively in the form of a meta-analysis due to the wide variability between the between individual observational studies, including different settings/country of residence, types of medicines, adherence measures, as well as the different clinical and social characteristics of the population. Also, in addition, different studies used different analytical approaches and adjusted for different confounders. 

We therefore accept that ascertaining specific reasons for variation in adherence is challenging, thus it would be inappropriate to provide a pooled quantitative value in terms of ethnicity for who is, or who is not, likely to be adherent to their antidiabetic medication. 

However, we also feel that ethnicity of a patient could still have an important role in a patient being adherent to their medicines. Rather than employ a meta-analytical approach, we were interested in establishing if there were differences in medicines adherence between ethnicities across a population.

Despite there being many primary research studies reporting this information, there has been no systematic summary of this information. As such, we decided a narrative synthesis approach would be the best method to meet our objectives.

We believe there is value in narratively synthesising what exists in the literature with describing the limitations and our view; it is clear that there is a demonstrable influence from ethnicity, albeit the exact quantification of the size of this influence may not be possible to measure, given the wide variability of the studies. 

Patients from minority ethnic populations, who are socioeconomically disadvantaged almost always have poorer outcomes that their white counterparts. This is something that needs to be urgently addressed within our healthcare systems and we believe this synthesis will help define some of these challenges. 

2-Authors should briefly define the various adherence measures. 

Thanks for the suggestion. We have now provided a summary definition of the various adherence measures as follows:

[Page:3, Line:79-89]:

"Medication adherence can be measured using direct or indirect measurements[1]. Direct measurements include measuring the concentration of the medicines, their metabolite and detection of a biological marker given with drug in bodily fluids such as blood or urine while observing the patient’s administration of their medications[1]. Indirect methods are the more popular in adherence research as the direct method is costly and challenging to perform[1]. It can be classified according to the World Health Organisation (WHO) into objective and subjective measures[2]. Subjective measurements include methods needing either patient self-report or health care professional evaluations of adherence to prescribed medicines[2]. On the other hand, objective measurements involve counting of pills, electronic monitoring, analysis of the secondary database[2]. "

3-How do individual studies (and your review) deal with mixed ethnicity? 

Thanks for your question.

For individual studies, only six reported details about dealing with people from mixed race/ethnicity [3-8], three of them included people from a mixed ethnicity[3, 5, 8]. Two of these studies reported mixed ethnicity based on the Office of National Statistics official UK ethnicity categories[5, 8] and the third included people from mixed ethnicity in Hawaii based on self-reported data from member surveys[3].

For the remaining three studies, they excluded people from mixed ethnicity from the analysis[4, 6, 7].

For this review:

Mixed ethnicity was reported according to what was reported in each individual study.

4-In the results section, the authors often report "other races" and in some cases (eg lines 253-255) a difference of some ethnic groups is reported without reporting the comparison group. In the results section, the authors often report "other races”, which should be reported in full. Furthermore, in some cases (e.g. page 16, lines 253-256), a difference of some ethnic groups is reported without mentioning the comparative group.

In the results section, the authors often report "other races"

Thank you for pointing this out.

We agree that "other races" should be reported in full. However, "other races" are reported in this review according to what is reported in each individual study.

in some cases (eg lines 253-255) a difference of some ethnic groups is reported without reporting the comparison group.

Thank you very much for the reminder. We have made revisions accordingly as follows:

[Page:16, Line:269]:

"Of these, seven studies indicated that people from Black, Hispanic, and Asian ethnicity were more likely to be non-adherent to antidiabetic medications than people of White ethnicity, and these findings were statistically significant in four studies. "

In some cases (e.g. page 16, lines 253-256), a difference of some ethnic groups is reported without mentioning the comparative group.

[Page:17, Line:296]:

"The authors found that the adherence varied significantly by ethnicity, and the odds of adherence were lower for people of Black ethnicity than those of White and Hispanic ethnicity. "

Also, we went through the results to make sure that the difference between some ethnic groups was reported with reporting the comparison group. Therefore, we have revised another sentence as follows:

[Page:17, Line:308-309]:

"Lopez and colleagues and Osborn and colleagues demonstrated a significant association between increased adherence to antidiabetic medications and White ethnicity compared to African American, Hispanic, Asian, and American Indian ethnic groups. "

Response to reviewer #2

Reviewer #2 comments 

1- The authors performed a systematic review investigating the adherence to glucose-lowering treatment in patients with diabetes according to different ethnicities.The results are interesting, and this overview is much needed, significantly contributing to current knowledge. 

Thank you very much for the supportive comment.

2- on page 6, the authors correctly state that they prefer the term "ethnicity" to "race". This might be preferable not only for semantic reason, but also because, giving that ethnicity encompasses those cultural and geographical characteristics that are more likely to account for differences in medication adherence. In other words, it is also more technically sound to use the term ethnicity in this context because it ideally allows to take more confounders into account. This should be better reported in the methods section. We very much appreciate this comment and agree that ethnicity encompasses cultural and geographical characteristics that are more likely to account for differences in medication adherence, we revised in the method section as follows:

[Page:6, Line:164-172]:

"While some included studies use the term 'race', we prefer the term 'ethnicity', which is defined in this review according to Senior and Bhopal 'implies one or more of the following: shared origins or social background; shared culture and traditions that are distinctive, maintained between generations, and lead to a sense of identity and group; and a common language or religious tradition [9].' The term of 'ethnicity’ is preferred as it encompasses cultural and geographical characteristics, which will allows to take more confounders into account that are more likely to account for differences in medication adherence. For the included studies, we used labels provided by the authors of the original studies. "

3-The discussion could might be better elaborated. The main differences between ethnicities are not summarized in a narrative statement in the discussion, although they are thoroughly reported in Table 1. 

Thank you for your suggestion. 

Due to the limitation of variation between included observational studies in terms of setting, sample size, type of drugs, number of centres, study design, adherence measures, inclusion and exclusion criteria, clinical and social characteristics of the population, comorbidities, adjusted confounders as highlighted by reviewer#1, we cannot pull the data from table 1 and give a summary of differences by each ethnic group, so we added an overall difference between ethnicities as follows:

[Page:18, Line:323-326]:

"This variation was statistically significant in 34 studies out of 41 included studies and was overall observed between people from ethnic minorities and the majority populations in each specific study setting. Moreover, it is reported when adherence was measured using various measurement methods. "

4-were there any important differences between glucose-lowering classes? Is there any factor that makes different ethnicities prefer oral antidiabetics rather than injectables? 

Thank you for this question. It would have been interesting to explore this aspect. However, in the case of our review, it is out of scope for our research question as we mainly focus on exploring whether general adherence to antidiabetic medication varied by ethnicity. This suggestion has been noted for future research ideas.

5- access to diabetes medication in the world cannot be ignored, please refer to the latest WHO report https://www.who.int/publications/i/item/9789241565257

Thank you for pointing this out. We agree that access to diabetes medication is an important determinant of adherence. However, the majority of studies (33 out of 41) in this review are retrospective database studies, which analyse the data of people who already have access to their antidiabetic medications. 

We do accept that this is something that needs to be investigated but would be part of a different review in our opinion. The question of ability to pay for medicines would also be very relevant in this space, especially in certain healthcare systems around the world 

6-Moreover, despite the lack of studies on type 1 diabetes, it is fundamental to better develop the discussion regarding the challenges of treatment adherence in type 1 vs type 2 patients. Although I understand that this is not the main focus of the paper, discussing adherence in type 1 diabetes cannot disregard the need of injectables, the difference in availability of glucose-monitoring system and a substantially different target population.

Consider for example citing "Hsin O, La Greca AM, Valenzuela J, Moine CT, Delamater A. Adherence and glycemic control among Hispanic youth with type 1 diabetes:role of family involvement and acculturation.J Pediatr Psychol. 2010;35(2):156-166" 

Thank you for this suggestion. 

It would be interesting to explore this aspect in future work. However, in our review, it seems slightly out of scope as we mainly focus on exploring whether adherence to antidiabetic medication varied by ethnicity in people with type 1 or type 2 diabetes. 

"Hsin O, La Greca AM, Valenzuela J, Moine CT, Delamater A. Adherence and glycemic control among Hispanic youth with type 1 diabetes: role of family involvement and acculturation .J Pediatr Psychol. 2010;35(2):156-166"

Thank you for giving this study as an example. However, this study did not meet the inclusion criteria for our systematic review, as it only included one ethnic group; the eligibility criteria for our systematic review specified that the study must have more than one ethnic group in order to reflect on comparisons. 

7-it is certainly true that the cause of these disparities should be better investigated in future research, however that is not the only necessity and is also already partly known. The authors should speculate on possible tools for identifying adherence issues and improving it across different ethnic groups. Thank you for this suggestion. We added it to discussion section as follows:

[Page:20, Line:374-376]:

"Therefore, developing a cross-culturally validated adherence measure in future work may help identify and improve adherence issues across different ethnic groups. "

References:

1. Lam WY, Fresco P. Medication adherence measures: an overview. BioMed research international. 2015;2015.

2. Organization WH. Adherence to long-term therapies: evidence for action: World Health Organization; 2003.

3. Taira DA, Seto BK, Davis JW, Seto TB, Landsittel D, Sumida WK. Examining Factors Associated With Nonadherence And Identifying Providers Caring For Nonadherent Subgroups. J Pharm Health Serv Res. 2017;8(4):247-53.

4. Lee R, Taira DA. PEER REVIEWED: Adherence to Oral Hypoglycemic Agents in Hawaii. Preventing chronic disease. 2005;2(2).

5. McGovern A, Hinton W, Calderara S, Munro N, Whyte M, de Lusignan S. A Class Comparison of Medication Persistence in People with Type 2 Diabetes: A Retrospective Observational Study. Diabetes Ther. 2018;9(1):229-42.

6. Sutton CX, Carpenter D-A, Sumida W, Taira D. 2016 Writing Contest Undergraduate Winner: The Relationship Between Medication Adherence and Total Healthcare Expenditures by Race/Ethnicity in Patients with Diabetes in Hawai ‘i. Hawai'i Journal of Medicine & Public Health. 2017;76(7):183.

7. Juarez DT, Tan C, Davis JW, Mau MM. Using quantile regression to assess disparities in medication adherence. American journal of health behavior. 2014;38(1):53-62.

8. Langley CA, Bush J. The Aston Medication Adherence Study: mapping the adherence patterns of an inner-city population. Int J Clin Pharm. 2014;36(1):202-11.

9. Senior PA, Bhopal R. Ethnicity as a variable in epidemiological research. Bmj. 1994;309(6950):327-30.

---

## [Decision Letter · Decision Letter 1]

5 Feb 2023

Ethnic disparities in medication adherence? A systematic review examining the association between ethnicity and antidiabetic medication adherence

PONE-D-22-18891R1

Dear Dr. Husband,

We’re pleased to inform you that your manuscript has been judged scientifically suitable for publication and will be formally accepted for publication once it meets all outstanding technical requirements.

Kind regards,

Dured Dardari, Ph.D

Academic Editor

PLOS ONE

Additional Editor Comments (optional):

Reviewers' comments:

Reviewer's Responses to Questions

**Comments to the Author**

1. If the authors have adequately addressed your comments raised in a previous round of review and you feel that this manuscript is now acceptable for publication, you may indicate that here to bypass the “Comments to the Author” section, enter your conflict of interest statement in the “Confidential to Editor” section, and submit your "Accept" recommendation.

Reviewer #1: All comments have been addressed

2. Is the manuscript technically sound, and do the data support the conclusions?

Reviewer #1: Yes

3. Has the statistical analysis been performed appropriately and rigorously? 

Reviewer #1: N/A

4. Have the authors made all data underlying the findings in their manuscript fully available?

Reviewer #1: Yes

5. Is the manuscript presented in an intelligible fashion and written in standard English?

Reviewer #1: Yes

6. Review Comments to the Author

Reviewer #1: I thank Authors for accepting and answering my questions. All my concerns has been well addressed.

While I believe the presentation of results with the above-mentioned problems may be confusing, this manuscript may bring to light the issue of ethnic disparities.

7. PLOS authors have the option to publish the peer review history of their article (what does this mean?). If published, this will include your full peer review and any attached files.

Reviewer #1: No

---

## [Editor Report · Acceptance letter]

10 Feb 2023

PONE-D-22-18891R1 

Ethnic disparities in medication adherence? A systematic review examining the association between ethnicity and antidiabetic medication adherence 

Dear Dr. Husband:

I'm pleased to inform you that your manuscript has been deemed suitable for publication in PLOS ONE. Congratulations! Your manuscript is now with our production department. 

Kind regards, 

on behalf of

Dr. Dured Dardari 

Academic Editor

PLOS ONE